# Impact of the COVID-19 Pandemic on Farm Households' Vulnerability to Multidimensional Poverty in Rural China

**Yuan Li Liu** [1,2,3,†], **Kai Zhu** [1], **Qi Yao Chen** [1,2,3,†], **Jing Li** [4], **Jin Cai** [5], **Tian He** [1,2,3] and **He Ping Liao** [1,2,3,*]

1 School of Geographical Sciences, Southwest University, Chongqing 400715, China; liuyuanli200605@126.com (Y.L.L.); huanyou11@163.com (K.Z.); chenqiyaocqy@sina.com (Q.Y.C.); hetian2014@163.com (T.H.)

2 Southwest University Center for Targeted Poverty Alleviation and Regional Development Assessment, Chongqing 400715, China

3 State Cultivation Base of Eco-Agriculture for Southwest Mountainous Land, Southwest University, Chongqing 400715, China

4 School of Economics and Management, Southwest University of Science and Technology, Mianyang 621000, China; lij2019@swust.edu.cn

5 College of Tourism and Land Resource, Chongqing Technology and Business University, Chongqing 400067, China; caijin2011@126.com

* Correspondence: liaohp@swu.edu.cn; Tel.: +86-138-0836-0066

† Yuan Li Liu and Qi Yao Chen contributed equally to this manuscript.

**Abstract:** The COVID-19 pandemic has significantly impacted the economy and livelihoods of people worldwide. To analyze the impact of the pandemic on material conditions, income levels, health conditions, industrial development and employment opportunities of farmers in China's rural areas, especially poor areas and explore whether farmers can achieve stable poverty eradication during the COVID-19 pandemic, we interviewed 2662 farm households in poverty-stricken areas of China and used the multidimensional poverty measurement model, three-step feasible generalized least squares and propensity score matching to analyze data. We achieved the following results. First, the overall level of multidimensional poverty vulnerability index (MPVI) of the surveyed households was low and the MPVI of each dimension varied significantly. The MPVI of households in the treated group was higher than that of the control group. Second, COVID-19 increased farm households' vulnerability to multidimensional poverty in poverty-stricken regions; MPVI increased by 27.9%. Third, COVID-19's impact on various dimensions differed: the greatest impact was on the vulnerability to health deprivation, followed by industrial development, employment and income deprivation. However, the pandemic slightly reduced the vulnerability to material deprivation. Finally, we proposed various measures in response to the impact of the pandemic to assist farm households in poverty-stricken areas.

**Keywords:** COVID-19; vulnerability to multidimensional poverty; poverty-stricken areas; impact; farm households

## 1. Introduction

The COVID-19 pandemic swept the globe in 2020. It is the largest public health event faced by the world since the Second World War [1] and has significantly impacted societies, economies and the livelihoods of people in China and worldwide [2,3]. As of 31 December 2020, more than 200 countries or regions globally have had confirmed cases of COVID-19; there are more than 100 million confirmed cases worldwide and a cumulative death toll of more than 2,000,000 (https://www.who.int (accessed on 31 December 2020). The numbers of confirmed cases and deaths are still on the rise. To curb the spread of the pandemic, most countries have adopted social isolation measures, which include road closures, local lockdowns, transportation control and border closures. Although the spread of the virus has been effectively curbed, these measures have also affected socioeconomic development, which

has led to the suspension or low-level growth of economic development in most countries. This may instigate a worldwide economic predicament and global recession [4]. The current year, 2020, is the target year for the completion of a moderately prosperous society in all respects and the achievement of poverty alleviation in China. To ensure the timely completion of all operations, the Chinese government is currently coordinating measures for pandemic prevention and control as well as poverty alleviation. However, farm households in poverty-stricken areas have a low capacity for coping with risk. Coupled with the impact of the COVID-19 pandemic, the social and economic development of rural areas has been hindered, the development of village collective industries has stagnated and the selling of agricultural and sideline products has been halted. It is difficult for rural migrant workers to find local employment or to move to other areas for work; thus, the income of farm households has decreased. This not only increases the farm households' vulnerability to poverty but may also cause certain farm households to fall into poverty. Therefore, analyzing the impact of the COVID-19 pandemic on farm households' vulnerability to multidimensional poverty in poverty-stricken areas of China is of great practical significance for the timely realization of poverty alleviation among poverty-stricken populations as well as the historical eradication of absolute poverty.

The academic community has always attached great importance to the impact of epidemics on society and the economy. The prevailing view from existing studies is that epidemic prevention and control conflicts with socioeconomic development because epidemics are highly dynamic and transmissible. Although the spread of epidemics can be prevented by controlling transportation and the flow of people [5,6], this paralysis of the transportation network restricts socioeconomic development. For instance, the interruption of logistics leads to a short supply of food and hinders the consumption of agricultural and sideline products [7], while social isolation leads to market downturn and lower house prices and sales numbers [8]. Prevention and control strategies have been proposed based on the characteristics of major epidemics, such as H1N1 and Ebola, to seek a balance between reducing the virus spread and not hindering economic development [9]. To further clarify the impact of epidemics on socioeconomic development, scholars used the Severe Acute Respiratory Syndrome (SARS) epidemic as an example. They utilized cross-correlation functions to analyze the relationships between the time series of SARS cases and deaths in Beijing and transport, cargo transportation, tourism, household consumption patterns and gross domestic product growth [10]. Additionally, some scholars believe that environmental protection should be enhanced during the prevention and control of COVID-19 [11] and that international and domestic tourism should be controlled [12] to reduce the spread of the disease.

Poverty is an issue that has long plagued the world and is a major concern affecting the development of human societies [13]. Scholars have proposed a multidimensional poverty measurement method based on the multidimensionality, region-specificity and dynamic characteristics of poverty [14,15]. A multidimensional poverty index system, which accounts for education, health and living standards, has been constructed [16,17]. Methods, such as the probit regression model [18], first order stochastic dominance method [19] and Back Propagation (BP) neural network model and regression analysis [20], have been utilized to comprehensively measure the extent of multidimensional poverty. In addition, the theory of spatial poverty has been employed to portray the characteristics of the spatial patterns of multidimensional poverty. Furthermore, emphasis should be placed on the multidimensional poverty problem in distinct populations by analyzing the poverty rate of the disabled the challenges young women face in securing work and housing [21] and the difficulties of vulnerable groups who are just above the poverty line [22]. It is believed that such populations face a higher degree of multidimensional poverty and hence, warrant more attention. Vulnerability to poverty has been analyzed using latent transition analysis [23] and multilevel models [24], whereby individual and household vulnerabilities to poverty have been linked to socioeconomic characteristics [25,26]. Specifically, Gloede et al. evaluated more than 4000 households in Thailand and Vietnam to measure the degree

of risk for household poverty [27], while Gallardo et al. adopted the multidimensional poverty vulnerability measurement approach based on mean risk behavior to analyze Chile's vulnerability to poverty using the multidimensional poverty vulnerability index (MPVI) [28]. Azeem et al. compared the ex-post poverty and the ex-ante vulnerability to poverty of households and considered that most vulnerable households could be accurately identified through ex-ante measures of vulnerability to poverty [29]. Omotoso et al. focused on the issue of child poverty and believed that vulnerable "non-poverty-stricken" children were more vulnerable to poverty than children living in chronic poverty [30].

A review of the literature shows that scholars have investigated the impact of epidemics on socioeconomic development. In particular, the impact of COVID-19 on transportation, tourism and agricultural and sideline products in China and in the world has been described. For example, the impact of the COVID-19 pandemic during the first quarter of 2020 is estimated to have resulted in a 3.11% reduction in the aggregate volume of agricultural production in Southeast Asia [31]. Further, measures for the prevention and control of the pandemic and those for economic development have also been proposed. The academic community has focused on poverty in rural areas and emphasized the measurement of multidimensional poverty, its influencing factors, characteristics of spatial patterns and the vulnerability to poverty in the income or finance dimensions. However, the relevant existing research leaves room for further investigation of at least two aspects. First, the impact of COVID-19 on poverty lacks micro perspectives and quantitative analyses. The existing research primarily focuses on the macro perspectives of the impact of the pandemic on socioeconomic development; quantitative analyses targeting farm households have not been conducted. Second, research on vulnerability to multidimensional poverty is still in its infancy—existing research on poverty vulnerability focuses primarily on a single dimension of household income or finance and the research methodology lacks diversity. The index system used for assessing vulnerability to poverty remains to be improved.

In view of the above, we reviewed and summarized relevant research and employed methods such as the multidimensional poverty measurement model, three-step feasible generalized least squares (FGLS) and propensity score matching (PSM) to measure the MPVI of 2662 farm households in poverty-stricken regions of China from a micro perspective. The impact of the COVID-19 pandemic on farm households' vulnerability to multidimensional poverty in such regions was analyzed. Finally, targeted solutions were proposed to provide a scientific reference for the Chinese government to achieve timely poverty alleviation in all populations living in poverty.

## 2. Materials and Methods

### 2.1. Data Source and Descriptive Analysis of the Indexes

#### 2.1.1. Data Source

This is shown in Figure 1, all data were collected in June 2020 by 88 investigators from our research group in four counties of the Inner Mongolia Autonomous Region that had been lifted out of poverty—Naiman Banner, Bairin Left Banner, Morin Dawa Daur Autonomous Banner and Xinghe County (http://www.nmg.gov.cn/ (accessed on 31 December 2020). The primary investigation methods included data collection, questionnaire surveys and semi-structured interviews. All subjects provided informed consent before participating in the study. The study was conducted in accordance with the Declaration of Helsinki and the protocol was approved by the Poverty Relief Office of Inner Mongolia.

First, multistage sampling was employed with reference to the literature [32]. Sample counties were randomly selected from the autonomous region; sample villages were randomly selected from each sample county; and sample households were randomly selected from each sample village. The population sizes of the sample villages were used to determine the number and list of households for sampling. Data collection and semi-structured interviews were conducted by the lead investigators. They conducted exchanges and interviews with at least five or more cadres to grasp the basic situation of the farmers in the village and understand the extent to which the farmers were affected by the pandemic. The Participatory

Rural Appraisal method was utilized, in which two investigators per group were assigned and completed a questionnaire survey using "Wenjuanxing" software (paid version). The geographic coordinates (longitude and latitude) of each household were recorded. Finally, two investigators the roles were exchanged and the data collected from the field surveys were examined and verified. Samples with errors or missing items were eliminated: Of the 3000 households that were sampled according to the predetermined protocols, an effective sample size of 2662 households were included (an effective rate of 88.73%). All data were processed with principal component analysis (PCA) for dimensionality reduction. The principal components of the poverty indexes were extracted to eliminate multicollinearity and to ensure the scientific and objective construction of the index system.

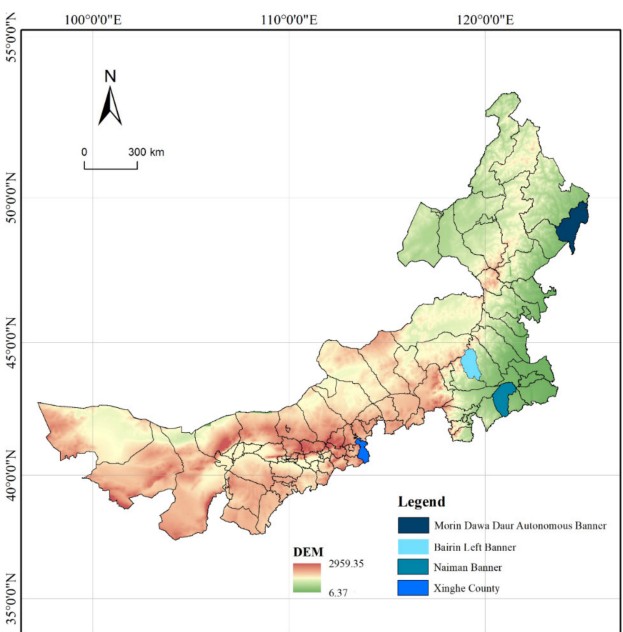

**Figure 1.** Geographical location of the study area.

### 2.1.2. Descriptive Analysis of the Indexes

The two main categories of the indexes used in this study were dependent variables and covariates. Considering dependent variables, we adopted the multidimensional poverty theory and utilized 15 indexes from five dimensions—material, income, health, employment and industrial development—to characterize the indexes for unidimensional and multidimensional poverty among farm households. This enabled the measurements of vulnerability to poverty in the five dimensions, as well as the vulnerability to multidimensional poverty Table 1. Regarding the covariates, we selected 11 indexes—including the type of farm household, the gender of the household head, dependency ratio, microfinance for poverty alleviation and communication facilities—to characterize the status of the farm households from the aspects of family characteristics, family burden and external conditions. The data were stratified into treated and control groups based on whether the farm households had been affected by the pandemic. Specifically, farm households in the treated group (sample size of 1572) indicated that they had been affected by the pandemic and those in the control group (sample size of 1090) indicated that they had not been affected by the pandemic. The data showed that the indexes of medical burden ratio, household size and property income were higher in the treated group than in the control group. The indexes of opportunities to participate in cooperative businesses, frequency of labor skill training, number of rural migrant workers, wage income, transportation conditions, communication facilities and microfinance for poverty alleviation were lower than that in the control group, with a decrease of more than 10%. These findings indicated that the farm households affected by the pandemic were impacted to a certain extent in relation to industrial development, skill training and labor migration.

**Table 1.** Variable description and descriptive statistics.

| Variabl Type | Variable Name | Variable Description | Control Group | | Treated Group | |
|---|---|---|---|---|---|---|
| | | | Mean | Standard Deviation | Mean | Standard Deviation |
| Quality of materials | Quality of safe housing | Brick + concrete house = 3; Brick + tile or brick + wood house = 2; Soil + wood house = 1 | 2.83 | 0.40 | 2.89 | 0.33 |
| | Safety of drinking water | Tap water = 4, Well water = 3, River water = 2, Other = 1 | 3.54 | 0.57 | 3.50 | 0.57 |
| | Average cultivated land area per household member | Cultivated land area/household size | 10.00 | 1.13 | 10.13 | 1.09 |
| Income level | Average net income per household member | Net household income/household size | 10,531.39 | 4493.89 | 10,133.78 | 4797.48 |
| | Wage income | Income from rural migrant workers in the household | 13,832.26 | 10,091.82 | 11,983.18 | 12,424.89 |
| | Property income | Property income from household land transfer and dividends and so forth. | 2954.60 | 2736.02 | 3449.52 | 3303.62 |
| Health status | Physical conditions | Healthy = 4, Chronic disease = 3, Serious illness = 2, Severe disability = 1 | 3.00 | 0.71 | 3.08 | 0.77 |
| | Medical burden ratio | Medical expenses/total household income | 1.24 | 0.51 | 3.54 | 0.80 |
| | Healthcare quality | Very good = 4, Good = 3, Average = 2, Poverty-stricken = 1 | 3.54 | 0.59 | 3.51 | 0.58 |
| Employment status | Labor skill training | Did not participate = 0, Participated = 1 | 0.92 | 0.27 | 0.39 | 0.49 |
| | Number of rural migrant workers | Number of rural migrant workers in the household | 1.14 | 0.86 | 0.83 | 0.77 |
| | Public welfare job | Number of household members with public welfare jobs | 0.56 | 0.52 | 0.61 | 0.69 |
| Industrial development | Industry support fund | With industry support fund = 1, Without industry support fund = 0 | 0.78 | 0.42 | 0.82 | 0.38 |
| | Industrial development outcome | Very good = 4, Good = 3, Average = 2, Poverty-stricken = 1 | 3.39 | 0.53 | 3.31 | 0.49 |
| | Participation in cooperative businesses | Participated = 1, Did not participate = 0 | 0.67 | 0.47 | 0.22 | 0.41 |
| Covariate | Type of farm household | General household = 1, Household lifted out of poverty = 2, Household with the minimum living guarantee = 3, Household with the five guarantees = 4 | 1.64 | 0.56 | 1.57 | 0.56 |
| | Household size | Number of people in the household | 2.32 | 1.06 | 2.78 | 1.19 |
| | Gender of household head | Male = 1, Female = 0 | 0.82 | 0.38 | 0.87 | 0.33 |
| | Education level of household head | University = 4, High school = 3, Junior high school = 2, Elementary school or below = 1 | 1.39 | 0.55 | 1.46 | 0.56 |
| | Ethnicity | Han = 1, Ethnic minority = 0 | 0.70 | 0.46 | 0.72 | 0.45 |
| | Number of people out of the labor force | Number of people in the household who are out of the labor force | 0.18 | 0.43 | 0.19 | 0.44 |
| | Dependency ratio | Number of elderly over 60 years and children under 16 years/household size | 1.16 | 0.85 | 1.04 | 0.91 |
| | Sanitary condition | Very good = 4, Good = 3, Average = 2, Poverty-stricken = 1 | 3.58 | 0.56 | 3.54 | 0.55 |
| | Microfinance for poverty alleviation | Applied for microfinance for poverty alleviation = 2, Did not apply = 1 | 0.29 | 0.45 | 0.18 | 0.49 |
| | Transportation conditions | Very convenient = 4, Convenient = 3, average = 2, Not convenient = 1 | 3.60 | 0.57 | 1.90 | 0.55 |
| | Communication facilities | Very good = 4, Good = 3, Average = 2, Poverty-stricken = 1 | 3.57 | 0.57 | 1.89 | 0.59 |

*2.2. Methodology*

2.2.1. Data Normalization

The indexes used in this study had different base units and included both positive and negative indexes. Therefore, min-max normalization was employed to normalize the raw data for each index as previously described [33,34], using the following formulas:

Positive index:

$$Y_{ij} = \frac{X_{ij} - X_{min}}{X_{max} - X_{min}} \tag{1}$$

Negative index:

$$Y_{ij} = \frac{X_{max} - X_{ij}}{X_{max} - X_{min}} \tag{2}$$

where $Y_{ij}$ is the normalized index value, $X_{ij}$ is the raw data of the *j*-th index of the *i*-th farm household in the surveyed area and $X_{max}$ and $X_{min}$ are the maximum and minimum values of the *j*-th index, respectively.

2.2.2. Multidimensional Poverty Vulnerability Model

Drawing on relevant existing research [35], we defined the poverty vulnerability indexes by expected poverty. We employed the three-step FGLS to estimate the probability of farm households falling into multidimensional poverty in the dimensions of material, income, health, employment and industrial development [18]. This is the MPVI of the farm households and is calculated using the following formula:

$$MPVI_{h,t} = Pr(D_{h,t+1} > Z) \tag{3}$$

In Equation (3), $MPVI_{h,t}$ represents the probability that the multidimensional poverty vulnerability index of a farm household at period t + 1 is higher than a multidimensional deprivation threshold, while $Z$ represents the multidimensional deprivation threshold. According to the literature [36], the unidimensional and multidimensional deprivation threshold is 1/3. As the multidimensional poverty index $D_{h,t+1}$ during period t + 1 is unknown, it can be expressed as a function of the observable $X_h$ and the error ei, which includes shock factors. To this end, the multidimensional poverty index can be expressed as follows:

$$D_h = X_h a_h + e_h \tag{4}$$

In Equation (4), $X_h$ represents the observable household characteristics. Variables selected in this study included the gender and education level of the household head and dependency ratio. $e_h$ is a perturbation value with a mean of 0 and a variance of $\sigma_{e,h}^2$. As $\sigma_{e,h}^2$ may not follow a normal distribution, it can be expressed as follows:

$$\sigma_{e,h}^2 = X_h \beta_h + \mu_h \tag{5}$$

Therefore, the three-step FGLS was used to estimate the expected value $\hat{E}$ and the variance $\hat{\sigma}_e^2$ of the farm households' MPVI. The formula for calculating the MPVI of farm households can thus be transformed into the following:

$$MPVI_{h,t} = Pr(D_{h,t+1} > Z) = \Phi\left(\frac{\hat{E} - Z}{\hat{\sigma}_h}\right) = \Phi\left(\frac{X_h \hat{\alpha}_{FGLS,h} - Z}{\sqrt{X_h \hat{\beta}_{FGLS,h}}}\right) \tag{6}$$

2.2.3. Estimation by Propensity Score Matching

The Propensity Score Matching (PSM) estimation method was first proposed by Rosenbaum and Rubin (1983). They believed that when assessing policy outcomes, a higher similarity between the control and treated groups indicated a lower sample selection bias, which results in a more reliable assessment of policy outcomes [37]. The advantage of PSM is its ability to transform a multivariate into an index; that is, a propensity score (PS) and

the PS value can be used for matching the control and treated groups. This effectively reduces self-selection and confounding bias and allows more reliable treatment effects to be obtained [38]. Therefore, we performed PSM to estimate the impact of COVID-19 on farm households' vulnerability to multidimensional poverty in poverty-stricken areas of China. PS was used as the probability of farm households being impacted by COVID-19. The basic steps of the PSM estimation method are as follows:

Step 1: Estimation of PS. The logit model was employed to calculate the conditional probability of each sample farm household to be affected by the COVID-19 pandemic [39] and the value of this probability is the PS value.

$$p(X_i) = Pr(D_i = 1 \mid X_i) = \frac{exp(? = X_i)}{1 + exp(? = X_i)} = E(D_i = 1 \mid X_i) \tag{7}$$

In Equation (7), $X_i$ is a series of factors that affect farm households' vulnerability to multidimensional poverty; it also serves as a covariate in the PS model. β is the corresponding estimated coefficient.

Step 2: Matching of PS. The farm households affected by COVID-19 were matched to each farm household with similar PS values, who were not affected by the pandemic in the control group. This ensured that the main characteristics of the control and treated groups were as similar as possible. Samples that could not be matched were eliminated. There are several matching approaches for PSM. We utilized three approaches commonly used in the literature [40]: nearest neighbor matching, radius matching and kernel matching.

Step 3: Assessment of matching quality. The balance requirement was assessed to determine whether statistically significant differences between the two groups persisted after "resampling." This would ensure that the matching procedures balanced the data and achieved the effect of a randomized experimental design.

Step 4: Calculation of average treatment effect (ATT). The ATT and implication of the COVID-19 pandemic on the control and treated groups after matching were compared [41].

$$\begin{aligned} ATT &= E[Y_{1i} - Y_{0i} \mid D_i = 1] = E\{E[Y_{1i} - Y_{0i} \mid D_i = 1, p(X_i)]\} \\ &= E\{E[Y_{1i} \mid D_i = 1, p(X_i)] - E[Y_{0i} \mid D_i = 0, p(X_i)] \mid D_i = 1\} \end{aligned} \tag{8}$$

In Equation (8), $Y_{1i}$ and $Y_{0i}$ represent the MPVI of the sample farm households in the treated group and the control group, respectively.

## 3. Results

*3.1. Measurement of Vulnerability to Multidimensional Poverty*

The vulnerability indexes of unidimensional and multidimensional poverty in farm households were measured using the MPVI model Table 2. Overall, the MPVI of farm households was low, ranging from 0.018 to 0.164 and the vulnerability indexes of each dimension of poverty varied significantly. Specifically, the vulnerability indexes in the employment and income dimensions were relatively high, with values at 0.439 and 0.265, respectively. Next were the vulnerability indexes in the industrial development and health dimensions, which had values of 0.086 and 0.030, respectively. The vulnerability index of the material dimension was the lowest at 0.006.

Since 2014, the Chinese government has aggressively implemented a precise poverty alleviation strategy. Large-scale investigations have been conducted on all farm households, on the principle that poverty-stricken people should be "free from worries over food and clothing and have access to compulsory education, basic medical services and safe housing." Farm households were guaranteed to enjoy the security of safe housing and drinking water. Therefore, the vulnerability index for material dimension was the lowest in the households surveyed in this study. Comparing the MPVI between groups, the MPVI of farm households in the treated group was 1.2 times that of the control group, with the vulnerability indexes in five dimensions of poverty being higher than those of the control group. Specifically, the unidimensional poverty vulnerability index with the largest difference between the

groups was the vulnerability index in the income dimension, which was 0.284 in the treated group—a 19.53% increase compared with the control group. The vulnerability indexes in the health and employment dimensions of the treated group increased by 13.61% and 12.94%, respectively, compared with the control group. Meanwhile, the difference in the vulnerability index in the industrial development dimension between the two groups was 0.006. The vulnerability index in the material dimension remained unchanged between the two groups. These results indicated that the treated group had higher vulnerabilities to both unidimensional and multidimensional poverty than the control group.

**Table 2.** Descriptive statistics of the measurements of farm households' vulnerability to multidimensional poverty in the surveyed areas.

| Dimension | Group | Maximum | Upper Quartile | Median | Lower Quartile | Minimum | Average | Standard Deviation | Coefficient of Variation |
|---|---|---|---|---|---|---|---|---|---|
| Vulnerability to material deprivation | Overall | 0.017 | 0.006 | 0.005 | 0.004 | 0.001 | 0.006 | 0.002 | 0.307 |
| | Control group | 0.017 | 0.006 | 0.005 | 0.005 | 0.001 | 0.006 | 0.002 | 0.301 |
| | Treated group | 0.016 | 0.006 | 0.005 | 0.004 | 0.001 | 0.006 | 0.002 | 0.310 |
| Vulnerability to income deprivation | Overall | 0.497 | 0.292 | 0.268 | 0.239 | 0.023 | 0.265 | 0.063 | 0.238 |
| | Control group | 0.395 | 0.280 | 0.256 | 0.211 | 0.023 | 0.237 | 0.065 | 0.274 |
| | Treated group | 0.497 | 0.301 | 0.274 | 0.254 | 0.172 | 0.283 | 0.054 | 0.189 |
| Vulnerability to health deprivation | Overall | 0.256 | 0.040 | 0.022 | 0.012 | 0.002 | 0.030 | 0.027 | 0.901 |
| | Control group | 0.256 | 0.036 | 0.018 | 0.010 | 0.003 | 0.027 | 0.027 | 0.973 |
| | Treated group | 0.250 | 0.042 | 0.024 | 0.012 | 0.002 | 0.031 | 0.027 | 0.854 |
| Vulnerability to employment deprivation | Overall | 0.691 | 0.466 | 0.446 | 0.414 | 0.048 | 0.439 | 0.077 | 0.176 |
| | Control group | 0.535 | 0.459 | 0.439 | 0.370 | 0.048 | 0.408 | 0.085 | 0.209 |
| | Treated group | 0.691 | 0.471 | 0.449 | 0.424 | 0.321 | 0.461 | 0.063 | 0.136 |
| Vulnerability to industrial development deprivation | Overall | 0.238 | 0.110 | 0.079 | 0.060 | 0.029 | 0.086 | 0.032 | 0.378 |
| | Control group | 0.199 | 0.105 | 0.074 | 0.060 | 0.029 | 0.082 | 0.029 | 0.360 |
| | Treated group | 0.238 | 0.113 | 0.083 | 0.059 | 0.031 | 0.088 | 0.034 | 0.386 |
| Vulnerability to multidimensional deprivation | Overall | 0.164 | 0.067 | 0.050 | 0.040 | 0.018 | 0.056 | 0.026 | 0.464 |
| | Control group | 0.159 | 0.058 | 0.049 | 0.038 | 0.018 | 0.051 | 0.017 | 0.345 |
| | Treated group | 0.164 | 0.072 | 0.050 | 0.042 | 0.018 | 0.060 | 0.030 | 0.499 |

To further eliminate the effects of the base units and scales of the index data, we calculated the coefficient of variation of the indexes to compare the variability of the dimensions. Table 2 shows that each dimension exhibits different degrees of heterogeneity. The level of dispersion in descending order is vulnerability to health deprivation, vulnerability to industrial development deprivation, vulnerability to material deprivation, vulnerability to income deprivation and vulnerability to employment deprivation.

*3.2. Analysis of the Impact of COVID-19 on Farm Households' Vulnerability to Multidimensional Poverty*

3.2.1. Propensity Score Matching and Assessment

Assessing Conformity to Balance Requirement

After the PS of the control and treated groups were matched, the balancing of data between the two groups was examined to determine whether significant differences existed. Using the nearest neighbor matching approach as an example, we examined whether the PSM results of the control and treated groups satisfied the balancing requirement. As shown in Table 3, the T-values of the 11 covariates were not significant and the absolute values of the standard deviations were within 10%. This indicated that the matching quality of the control and treated groups was satisfactory. Further, except for the dependency ratio, the biases of the covariates had varying degrees of reduction. It was evident that the control and treated groups did not exhibit statistical differences after matching and the effect of a randomized experiment was achieved [41].

**Table 3.** Balance requirement assessment after propensity score matching (nearest neighbor matching).

| Variable | Mean | | T-Test | | V(T)/ | |
|---|---|---|---|---|---|---|
| | Treated Group | Control Group | %bias | T | P > T | V(C) |
| Dependency ratio | 0.26531 | 0.2585 | 3.1 | 0.51 | 0.608 | 1.08 |
| Number of people out of the labor force | 0.06037 | 0.06675 | −5.9 | −0.88 | 0.38 | 0.85 |
| Sanitary condition | 0.12798 | 0.13265 | −3.4 | −0.55 | 0.584 | 0.88 |
| Gender of household head | 0.17347 | 0.18707 | −3.8 | −0.61 | 0.544 | |
| Ethnicity of household head | 0.35544 | 0.38435 | −6.4 | −1.03 | 0.305 | |
| Education level of household head | 0.83787 | 0.84297 | −2.8 | −0.45 | 0.652 | 0.94 |
| Household size | 0.25624 | 0.27239 | −8.6 | −1.39 | 0.166 | 0.92 |
| Type of farm household | 0.19501 | 0.20011 | −2.7 | −0.45 | 0.656 | 0.97 |
| Microfinance for poverty alleviation | 0.63435 | 0.61224 | 4.7 | 0.78 | 0.434 | |
| Convenience in transportation | 0.1559 | 0.1695 | −7.2 | −1.15 | 0.248 | 0.82 |
| Communication facilities | 0.17687 | 0.18141 | −2.3 | −0.38 | 0.702 | 0.88 |

Assessing Matching Quality

Nearest neighbor matching, radius matching and kernel matching were employed to compare the differences in the vulnerability to multidimensional poverty of the control and treated groups of farm households based on their PS. The robustness was validated (Table 4). The results showed that the Pseudo-*R2* values after matching were significantly reduced, compared with the values before matching. Specifically, the Pseudo-*R2* values after nearest neighbor matching, radius matching and kernel matching were reduced to 0.004, 0.003 and 0.002, respectively. This indicated that systematic differences of the variables were eliminated after matching.

**Table 4.** Assessment of matching quality with different matching approaches.

| Matching Approach | Quality Indicator | Matching Quality |
|---|---|---|
| Before matching | Pseudo-*R2* | 0.034 |
| | Average standardized bias | 13.93 |
| | *T*-test | 44.68 |
| Nearest neighbor matching | Pseudo-*R2* | 0.004 |
| | Average standardized bias | 2.82 |
| | *T*-test | 3.57 |
| Radius matching (0.005) | Pseudo-*R2* | 0.003 |
| | Average standardized bias | 4.02 |
| | *T*-test | 9.44 |
| Kernel matching (0.005) | Pseudo-*R2* | 0.002 |
| | Average standardized bias | 4.12 |
| | *T*-test | 10.82 |

The average standardized bias of the variables before matching was 13.93, while the values were significantly reduced after matching using either one of the three matching approaches. Specifically, the average standardized bias after nearest neighbor matching exhibited the largest reduction: 79.76%. The average standardized biases after radius matching and kernel matching were reduced by 71.14% and 70.42%, respectively. The reduction of the average standardized bias after matching indicated that the matching procedures were well suited for the sample characteristics of the control and treated groups.

The *T*-test result for the variables before matching was 44.68 and the results after applying the three different matching approaches were reduced to 3.57, 9.44 and 10.82, respectively. The *T*-test results exhibited a significant decreasing trend. Smaller *T*-values indicated smaller differences in the average values of the variables between the control and treated groups after matching, which implied better matching quality.

To directly compare the effects before and after the matching in the control and treated groups, we used kernel density plots to compare and analyze the effects of PSM (*p* score)

using nearest neighbor matching, radius matching and kernel matching. Figure 2 shows pronounced differences in the distribution of PS score before matching and the area of common support is small. The distribution of the PS score in the control and treated groups becomes closer after using the three matching approaches and the areas of common support are markedly increased. This shows that the sample characteristics of the two groups are highly similar, which supports earlier results regarding the excellent matching quality of the control and treated groups.

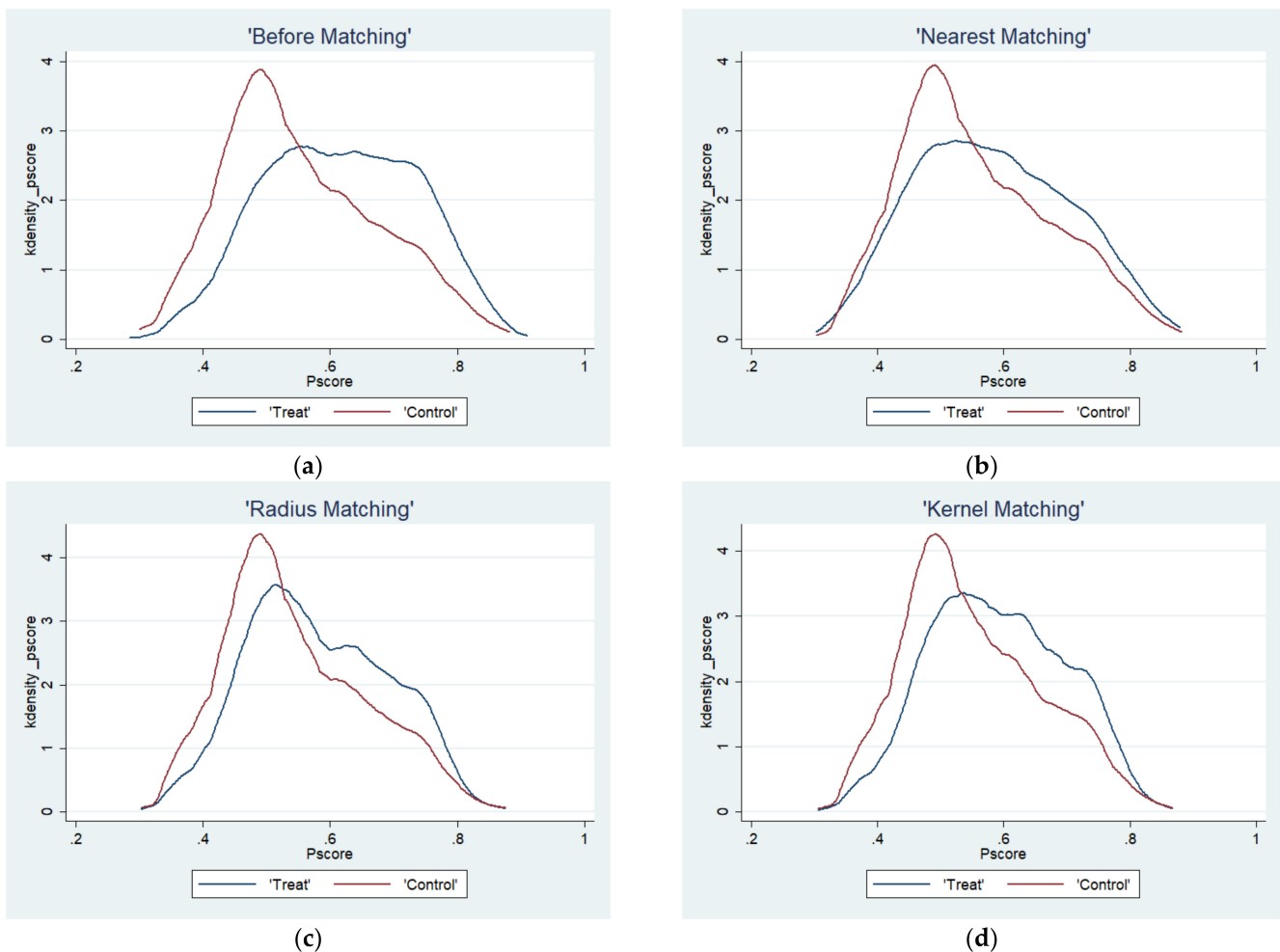

**Figure 2.** Distribution of kernel density before and after matching between control and treated groups: (**a**) Before Matching; (**b**) Nearest Matching; (**c**) Radius Matching; (**d**) Kernel Matching.

### 3.2.2. Analysis of ATT

We adopted the ATT method to measure the extent of the impact of COVID-19 on farm households' vulnerability to multidimensional poverty. We computed the ATT of the treated and control group samples by using nearest neighbor matching, radius matching and kernel matching. The average values obtained from the three approaches were then considered as the differences between the control and treated groups.

(1) The impact of COVID-19 on vulnerability to multidimensional poverty

Overall, the COVID-19 pandemic increased farm households' vulnerability to multidimensional poverty and the MPVI of farm households affected by the pandemic increased by 27.9%. The main reason is that some peasant households have deep multidimensional poverty. Although they have risen out of absolute poverty, their income is low, the income structure is unreasonable, the development of agricultural industry is slow, the stability of poverty alleviation is poor and the new pandemic has a significant impact on poverty.

Specifically, the ATT of COVID-19 on the vulnerability to multidimensional poverty measured using the three different matching approaches were all significant at a 1% level. The ATT measured by nearest neighbor matching, radius matching and kernel matching increased to 0.125, 0.129 and 0.129, respectively. The results showed that compared with those not affected by the pandemic, the MPVI of farm households affected by the pandemic increased by 27.48%, 28.10% and 28.12%, respectively. Moreover, the similarities in the results obtained by the three approaches indicated the robustness of the measured results Table 5.

**Table 5.** The impact of COVID-19 on vulnerability to multidimensional poverty.

| Variable Sample | Matching Approach | Treated Group | Control Group | ATT | S.E. | *T*-Stat |
|---|---|---|---|---|---|---|
| Vulnerability to material deprivation | Nearest neighbor matching (1:1) | 0.098 | 0.108 | −0.010 | 0.006 | −1.63 * |
| | Kernel matching | 0.106 | 0.115 | −0.010 | 0.007 | −1.49 |
| | Radius matching (caliper = 0.1) | 0.108 | 0.116 | −0.008 | 0.007 | −1.19 |
| Vulnerability to income deprivation | Nearest neighbor matching (1:1) | 0.603 | 0.466 | 0.137 | 0.009 | 16.01 *** |
| | Kernel matching | 0.620 | 0.471 | 0.149 | 0.008 | 18.48 *** |
| | Radius matching (caliper = 0.1) | 0.624 | 0.474 | 0.150 | 0.009 | 16.96 *** |
| Vulnerability to health deprivation | Nearest neighbor matching (1:1) | 0.355 | 0.226 | 0.129 | 0.007 | 19.64 *** |
| | Kernel matching | 0.345 | 0.216 | 0.128 | 0.006 | 21.19 *** |
| | Radius matching (caliper = 0.1) | 0.336 | 0.208 | 0.128 | 0.006 | 20.26 *** |
| Vulnerability to employment deprivation | Nearest neighbor matching (1:1) | 0.764 | 0.547 | 0.217 | 0.010 | 21.82 *** |
| | Kernel matching | 0.765 | 0.553 | 0.212 | 0.008 | 25.27 *** |
| | Radius matching (caliper = 0.1) | 0.771 | 0.553 | 0.218 | 0.009 | 23.6 *** |
| Vulnerability to industrial development deprivation | Nearest neighbor matching (1:1) | 0.385 | 0.254 | 0.131 | 0.011 | 11.97 *** |
| | Kernel matching | 0.393 | 0.250 | 0.142 | 0.010 | 13.7 *** |
| | Radius matching (caliper = 0.1) | 0.390 | 0.255 | 0.135 | 0.011 | 12.15 *** |
| Vulnerability to multidimensional deprivation | Nearest neighbor matching (1:1) | 0.455 | 0.330 | 0.125 | 0.004 | 30.17 *** |
| | Kernel matching | 0.460 | 0.331 | 0.129 | 0.004 | 34.82 *** |
| | Radius matching (caliper = 0.1) | 0.460 | 0.331 | 0.129 | 0.004 | 32.44 *** |

Note: *, *** denote significance at the 10%, 1% levels, respectively.

(2) The impact of COVID-19 on vulnerability to each dimension of poverty

Vulnerability to poverty in the material dimension. The results showed that the measurements using nearest neighbor matching were significant and that using the other two approaches was not significant. In addition, the ATT values decreased with all three approaches, indicating that the pandemic reduced the vulnerability to poverty in the material dimension. The main reason was that the vulnerability index in the material dimension was the lowest in the control and treated groups before matching, with an average value of 0.006. This suggested that the two groups of farm households had received basic security to access safe housing, drinking water and cultivated land area per household member. Therefore, the pandemic had no effect on the vulnerability of farm households in the material dimension. Rather, it reinforced the access to quality housing and safe drinking water, resulting in a slight increase in the ATT of the treated group.

Vulnerability to poverty in the income dimension. The results of the three matching approaches were all significant at the 1% level. In addition, the ATT of the treated group was 0.145 higher than that of the control group. The pandemic led to a 23.58% increase in the vulnerability index of the income dimension in farm households of the treated group, indicating that the COVID-19 pandemic had a great impact on the income of poverty-stricken farm households. Quarantine measures were imposed in most parts of China since the beginning of the pandemic, which hindered the movement of rural migrant workers and led to a significant drop in their income. The average income from rural migrant workers from farm households in the treated group had reduced 1849.08 Yuan, compared with those in the control group. Notably, the net income per household member in the two groups did not differ significantly, principally due to the higher transferable income received by the treated group. Because the study area is a deeply impoverished

area, in order to ensure they would not fall into poverty due to the pandemic, the Chinese government increased the financial subsidies to farmers affected by the pandemic.

Vulnerability to poverty in the health dimension. The results of the three matching approaches were all significant at the 1% level and the ATT of the treated group increased by 0.128, compared with the control group. The vulnerability index in the health dimension increased by 37.20%, which was the largest increase among all dimensions, indicating that the COVID-19 pandemic had an extremely pronounced impact on farmers' health. The main reason was due to the inconvenience of accessing medical services experienced by farmers during the pandemic. The proportion of farmers who chose to visit a physician due to a cold or fever became especially low and they could only be isolated at home. In addition, the elderly suffered from frequent episodes of chronic diseases in winter. They were the group most impacted by the pandemic, due to their poor physical conditions and difficulty accessing quality medical services.

Vulnerability to poverty in the employment dimension. Employment is the key to poverty alleviation for farm households in poverty-stricken areas and an effective means of increasing the income of farm households. The results of the three matching approaches were all significant at the 1% level. In addition, the ATT of the treated group increased by 0.216, compared with the control group. The vulnerability index in the employment dimension increased by 28.35%, indicating that the pandemic had a significant impact on vulnerability to employment deprivation in farm households. During the pandemic, the provision of labor skill training and the number of rural migrant workers were severely affected in the surveyed areas. Rural migrant workers were temporarily unemployed and labor skill training was suspended, due to the banning of gatherings, resulting in higher vulnerability to employment deprivation.

Vulnerability to poverty in the industrial development dimension. Industrial development is an important channel to achieve steady poverty alleviation. It is also an important means for the poor to achieve a moderately prosperous life. The results of the three matching approaches in the control and treated groups were all significant at the 1% level. However, the ATT of the treated group increased by 0.136 and the vulnerability index of the treated group in the industrial development dimension increased by 34.96%. This increase was only second to the increase in the health dimension. The main reason was that farm households affected by the pandemic had fewer opportunities to participate in cooperative businesses, which hindered industrial development. Poultry and livestock industries had been considerably affected, leading to poverty-stricken development progress and a significant reduction in farmers' satisfaction.

## 4. Conclusions and Suggestions

### 4.1. Conclusions

We collected evidence from 2662 farm households in poverty-stricken areas of China and employed the multidimensional poverty measurement model, the three-step FGLS and PSM to measure farm households' vulnerability to multidimensional poverty from a micro perspective. The impact of the COVID-19 pandemic on the vulnerability to multidimensional poverty among farm households in poverty-stricken areas of China was analyzed to provide theoretical support for the timely achievement of poverty alleviation in all populations living in poverty. The following conclusions can be drawn from the study:

The overall level of MPVI of farm households in the surveyed areas was low and the MPVI of each dimension varied significantly. In addition, the MPVI of the farm households in the treated group was significantly higher than that of the control group. The MPVI of farm households in the surveyed area was between 0.018 and 0.164. Specifically, the vulnerability indexes in the employment and income dimensions were the highest, followed by those in the industrial development and health dimensions. The vulnerability index in the material dimension was the lowest. Further, the MPVI of farm households in the treated group was 1.2–fold of that of the control group, with the vulnerability indexes in five dimensions of poverty being higher than those of the control group. In particular, the

vulnerability indexes in the income, health and employment dimensions exceeded that of the control group by more than 10%.

The balance requirement and matching quality of the control and treated groups were assessed after matching. Specifically, the T-values of all covariates were not significant and the absolute values of the standard deviations were within 10%, indicating that the matching quality of the control and treated groups was satisfactory. Further, the differences in the vulnerabilities to multidimensional poverty of the two groups of farm households were compared using nearest neighbor matching, radius matching and kernel matching. The Pseudo-R2, average standardized biases and T-values were significantly reduced after matching, indicating that the matching procedures eliminated the systematic differences of the variables and were well suited for the sample characteristics of the control and treated groups, which satisfied the study requirements.

The COVID-19 pandemic increased farm households' vulnerability to multidimensional poverty in poverty-stricken areas in China, although the effects on the vulnerabilities in different dimensions varied. Regarding the vulnerability to multidimensional poverty, the MPVI of farm households affected by the pandemic increased by 27.9%, indicating a greater impact on these households, compared with the control group. When considering the vulnerability in each dimension, the pandemic had the greatest impact on the vulnerability to health deprivation, as the vulnerability index increased by 37.2%. This was followed by vulnerabilities to industrial development, employment and income deprivations: The vulnerability indexes increased by 34.96%, 28.12% and 23.58%, respectively. However, the pandemic did not affect the vulnerability to material deprivation. Contrarily, it reduced the level of vulnerability to poverty in the material dimension.

### 4.2. Discussion

The 2020 COVID-19 pandemic has had a severe impact on global socio-economic development, making it highly difficult for countries to achieve the target of poverty eradication set out in the United Nations 2030 Sustainable Development Goals. To ensure sustainable development of the poverty-stricken population, the Chinese government adopted a series of measures to stabilize poverty alleviation and prevent this population from falling back into poverty; these measures were also aimed to prevent farmers from becoming poverty-stricken. Therefore, exploring the impact of the outbreak of the COVID-19 pandemic on the multidimensional poverty vulnerability of rural households in poor areas of China is a topic that has theoretical and practical significance.

The results of the study reveal that the overall multidimensional poverty vulnerability index of farmers in the study area is low, indicating that the implementation of poverty alleviation measures has been highly effective and the risk of non-poor people falling into poverty is low. Among the population studied, rural households were found to have the most abundant material conditions, with rural industries developing steadily; although rural health was found to be relatively high, abilities are lacking and the level of education and employment skills are low. Thus, this population is not stable enough to work outside their region and their family income structure is simple. Sustainability needs to be strengthened and it is a key factor affecting poverty vulnerability. After the outbreak of the COVID-19 pandemic in 2020, the material resources owned by farmers increased rather than decreased, with the help and care offered by the government. However, the physical health of farmers was affected because of social isolation caused by road closures and traffic controls, which made it difficult for farmers to seek medical treatment outside their areas. Besides, rural medical facilities were not geared to treat diseases other than COVID-19 during the spread of the infection, thus increasing the vulnerability of farmers to health poverty. Development of rural industry was hindered to a certain extent and agricultural spring production lacked supply of production materials; further, circulation of agricultural goods was poor with passive consumer demand and rising prices of raw materials, all of which are similar to findings from existing research. At the same time, the impact of the COVID-19 pandemic led to unemployment among migrant workers, whose income from

employment reduced significantly. They had to depend on employment opportunities organized by the government or by public welfare institutions. The pandemic therefore had a significant effect on peasant household multidimensional poverty vulnerability, increased the vulnerability and risk of the poor to fall back into poverty but the timely measures by the Chinese government were able to contain the risk of poverty as the poor population managed to emerge out of poverty in 2020. Thus, China was able to achieve the target of poverty eradication set out in the United Nations 2030 Sustainable Development Goals, ten years ahead of schedule.

The research methods adopted included the perspective of farmers, the integrated use of the multidimensional poverty measurement model, a three-step feasible generalized least squares and propensity score matching for the quantitative analysis. Through field research and several rounds of discussion with village cadres, we obtained basic information about the situation of farmers and conducted qualitative research. A novel aspect of our research method is that we used a combination of quantitative analysis and qualitative analysis. At the same time, our research was focused on the micro level to analyze the impact of the COVID-19 epidemic on poverty from the perspective of farmers in China's poor regions to gain insights into those who are at risk of falling into poverty. Research focusing on how the pandemic affects the rural poor is vital in studies on infectious disease and its spread.

However, our research project also has some data limitations; for instance, field survey was conducted only once and the peasant household data were lacking because data were static cross-section data without dynamic panel data and there was no continuous observation. Therefore, a comprehensive and persistent measurement of the impact of the pandemic on the vulnerability of peasant households to multidimensional poverty was not possible. The multidimensional poverty vulnerability evaluation index system needs to be developed further by fully considering the characteristics of farmers, endogenous motivation and national policy support. At the same time, owing to the lack of microcosmic data regarding the COVID-19 pandemic situation in each administrative village, the research results cannot be directly displayed from a spatial perspective. This point needs to be addressed in future research.

### 4.3. Suggestions

Based on the above conclusion and discussion, to resolve the impact of the COVID-19 pandemic on farm households' vulnerability to multidimensional poverty, especially the impact on employment, industrial development and incomes of rural migrant workers in poverty-stricken areas, we propose the following:

Intensify re-employment training for rural migrant workers to aid them in obtaining local employment. The unemployment issue among rural migrant workers is comparatively more severe because some are unable to travel due to the pandemic. Therefore, government should further intensify re-employment training for rural migrant workers in accordance with the needs of enterprises and the workers themselves. In addition, they should rely on local agro-industrial parks, industrial parks and poverty alleviation workshops to solve the unemployment issue among rural migrant workers.

Continue to increase policy-based loan support and aggressively develop distinctive industries. During the pandemic, some leading enterprises and cooperatives experienced capital chain rupture, which hindered industrial development. Therefore, the government should further increase financial support, lower loan thresholds accordingly, simplify loan procedures, shorten loan life cycles and offer discount loans to support leading enterprises and cooperatives and reduce their burden on business. Further, the government should direct enterprises to aggressively develop distinctive industries, such as modern agriculture and tourism, to drive the development of poverty-stricken farm households' livelihoods.

Promote an ongoing increase in farmers' income through multiple channels. Different authorities—including the Ministry of Health, the Ministry of Human Resources and Social Security, the Employment Bureau and the Bureau of Civil Affairs—should collaborate to introduce preferential policies and encourage farmers to innovate and start

their own businesses to drive their motivations. The implementation of civil assistance and minimum-security policies should be strengthened to ensure that funds can reach individual households, which will increase the transferable income of farm households. The government, society and enterprises should join forces to integrate funds for increasing public welfare positions, such as village cleaners, river administrators and road guards, which will increase the wage income of farm households.

**Author Contributions:** Y.L.L. and Q.Y.C. designed this study. J.L., J.C., T.H., K.Z. and Q.Y.C. conducted the investigation. Y.L.L. analyzed the data and wrote the manuscript. H.P.L., Y.L.L., T.H. and Q.Y.C. interpreted the results and reviewed the manuscript. All authors have read and agreed to the published version of the manuscript.

**Funding:** This research was supported by the research fund of Southwest University of Science and Technology (No. 19sx7106), the National Social Science Foundation of China (No. 20BSH079) and the Humanities and Social Sciences Project of the Ministry of Education of China (No. 20XJCZHOO5).

**Institutional Review Board Statement:** Not applicable.

**Informed Consent Statement:** Not applicable.

**Data Availability Statement:** All data, models, and code generated or used during the study appear in the submitted article.

**Acknowledgments:** We would like to thank the farmers and employees who work in government departments in Naiman Banner, Bairin Left Banner, Morin Dawa Daur Autonomous Banner and Xinghe County for their support and valuable data. We also acknowledge the financial support from the research group. Special thanks for the professional English editing service provided by Editage.

**Conflicts of Interest:** The authors declare no conflict of interest.

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
