# Peer review of "Impact of the COVID-19 Pandemic on Farm Households’ Vulnerability to Multidimensional Poverty in Rural China"

_sustainability, doi:10.3390/su13041842_

Round 1

Reviewer 1 Report

I have a problem with "control group". In my opinion, it's not really a control group as mentionned. It's a comparison between two different groups.

Author Response

Dear Editor:

        Thank you very much for giving me the opportunity to revise this paper. My team and I made detailed modifications and added explanations to the article in strict accordance with your  opinions. We provided responses to each of the reviewers. The details are as follows:

        According to the measurement method of propensity score matching (PSM), this study uses 11 covariables to select two different groups, one representing the situation before the pandemic (control group) and the other representing the situation after the pandemic (treatment group), so as to obtain the impact of the pandemic on multidimensional poverty vulnerability. After the discussion with my team, we still want to use the term "control group." If the reviewer still thinks there are drawbacks, we will revise it again according to the reviewer’s advice.

Reviewer 2 Report

The paper is well develloped and the methods of research are suitable for this paper. Acctually is rather a very techincal paper than a socio-economic one. For authors is rather important to develop the econometric part than to deepen the problems from the rural areas in the Covid 19 context. I would rather start the research from the suggestions and to intereact with the people /authorities and identify practical ways in offering solutions to better pass this period of time. 

Some ideas for authors:

  • to make a map of the affected areas and to colour in several degrees the higher or the less impact of Covid in these regions
  • to find why some regions were more affected than others in economic and social terms
  • to interact in real terms with the people from the rural areas, not only based on questionarry  

Author Response

Dear Editor:

Thank you very much for giving me the opportunity to revise this paper. My team and I made detailed modifications and added explanations to the article in strict accordance with your opinions.The details are as follows:

1Comments and Suggestions

To find why some regions were more affected than others in economic and social terms

Description of the modification:

I carefully reviewed the article and highlighted the reasons why society and economy are affected by the pandemic. For example, although some poor households have been lifted out of poverty, their income, housing, and medical care are not sufficient, leading to a significant increase in their vulnerability to multi-dimensional poverty after being affected by the outside world. Please refer to the red text in the manuscript for details.

2Comments and Suggestions

To interact in real terms with the people from the rural areas, not only based on questionary

Description of the modification:

According to the reviewer’s advice, I elaborated on the source of data for the paper, including not only the survey questionnaire, but also the communication and interviews with the local village cadres. Additionally, in the analysis of the impact of the pandemic on multidimensional poverty vulnerability, I added content of real interaction with farmers. Please refer to the red text in the manuscript for details.

3Comments and Suggestions

To make a map of the affected areas and to colour in several degrees the higher or the less impact of Covid in these regions.

Description of the modification:

Thanks to the reviewer for this advice. I would like to make a map of the affected areas, including the distribution of 2,662 samples from 20 villages in four counties, but there is not enough data for each village. Therefore, this study explored the situation of farmers' families affected by the pandemic to illustrate the characteristics and perspectives of farmers. It is a pity that there is no data for each administrative village, but it is a direction for our future research.

Reviewer 3 Report

The Authors of the article took up the current research problem. The research is well documented and the statistical analyzes are clear. The conclusions were also presented in a clear manner, adequate to the problem undertaken. However, correction requires:

  • formulation of the problem, research goal.

Currently in the text it has been formulated as follows: “To analyze the pandemic’s impact on households’ vulnerability to poverty, we interviewed 2,662 farm households in poverty-stricken areas of China and used the multidimensional poverty measurement model, three-step feasible generalized least squares, and propensity score matching to analyze data.” - analysis is not the aim of research, it is a way of solving the problem. Research questions should be clarified.

  • presentation of the research area - it is advisable to supplement with a map presenting the research area;
  • presentation of research results - it is worth deepening the analysis by showing the spatial differentiation of research results.

I encourage the Authors to improve the text. Its publication will be an interesting base for discussions on the international arena.

Author Response

Dear Editor:

Thank you very much for giving me the opportunity to revise this paper. My team and I made detailed modifications and added explanations to the article in strict accordance with your opinions. The details are as follows:

1Comments and Suggestions

Are the research design, questions, hypotheses and methods clearly stated? Formulation of the problem, research goal.

Description on the modification:

Based on the reviewer’s suggestion, I reviewed the research purpose and research method of this paper again, and elaborated it in detail: “To analyze the impact of the pandemic on material conditions, income levels, health conditions, industrial development, and employment opportunities of farmers in China's rural areas, especially poor areas, and explore whether farmers can achieve stable poverty eradication during the COVID-19 pandemic.” Please refer to the red text in the manuscript for details.

(2Comments and Suggestions

Presentation of the research area - it is advisable to supplement with a map presenting the research area.

Description of the modification:

Based on the reviewer’s suggestions, I added a geographical location distribution map of the research area, including the geographical locations of the four counties.

3Comments and Suggestions

Presentation of research results - it is worth deepening the analysis by showing the spatial differentiation of research results.

Description on the modification:

   The reviewer’s suggestion is very good. In Section 3.1 of this paper, the distribution of multidimensional poverty vulnerability level of farmers has been added in tabular form.

4Comments and Suggestions

I encourage the Authors to improve the text.

Description on the modification:

   In accordance with the reviewer’s advice, I invited a professional organization to polish the language of this paper again in order to meet the reviewer’s requirements.